# Deciphering Plant-Induced Responses toward *Botrytis cinerea* and *Plasmopara viticola* Attacks in Two Grapevine Cultivars Colonized by the Root Biocontrol Oomycete, *Pythium oligandrum*

**DOI:** 10.3390/jof9050511

**Published:** 2023-04-25

**Authors:** Amira Yacoub, Rana Haidar, Ouiza Mesguida, Jonathan Gerbore, Maya Hachicha, Eléonore Attard, Rémy Guyoneaud, Patrice Rey

**Affiliations:** 1E2S UPPA, CNRS, IPREM, Université de Pau et des Pays de l’Adour, 64000 Pau, France; 2INRAE, UMR1065, Santé et Agroécologie du Vignoble (SAVE), ISVV, 33883 Villenave d’Ornon, France; 3GreenCell, Biopôle Clermont-Limagne, 63360 Saint Beauzire, France

**Keywords:** grapevine, biocontrol, oomycete, cultivars, induction of resistance

## Abstract

Two major diseases that affect grapevine leaves and berries are controlled by the oomycete *Pythium oligandrum*. As the efficacy of biocontrol agents strongly depends on factors such as the trophic behaviors of pathogens and cultivar susceptibility, a two-disease approach was implemented to evaluate the activity of *P. oligandrum* against *Botrytis cinerea* (the necrotrophic fungus of gray mold) and *Plasmopara viticola* (the biotrophic oomycete of downy mildew) on two grapevine cultivars with different susceptibilities to these two pathogens. The results show that grapevine root inoculation with *P. oligandrum* significantly reduced *P. viticola* and *B. cinerea* infection on the leaves of the two cultivars, but with differences. This was observed when the relative expression of 10 genes was measured in response to each pathogen, and could be attributed to their lifestyles, i.e., biotrophic or necrotrophic, which are related to the activation of specific metabolic pathways of the plant. In response to *P. viticola* infection, genes from the jasmonate and ethylene pathways were mainly induced, whereas for *B. cinerea*, the genes induced were those of the ethylene–jasmonate pathway. The different levels of defense against *B. cinerea* and *P. viticola* could also explain the difference in cultivar susceptibility to these pathogens.

## 1. Introduction

Gray mold and downy mildew are major diseases that affect the berries and leaves of grapevines, causing serious economic losses in yield. Gray mold is caused by the necrotrophic plant pathogen *Botrytis cinerea* (Pers.), which kills grapevine cells before the colonization of plant tissues, while downy mildew is caused by the biotrophic oomycete *Plasmopara viticola* (Berk. & M. A. Curtis) Berl. & de Toni., which colonizes living plant cells [1]. Management of these diseases in vineyards mainly relies on preventive fungicide application, which may favor the development of pathogen-resistant strains [2], as well as the persistence of chemical residue in grapevine berries and wine [3]. In addition, due to the human health and environmental issues associated with the use of fungicides in vineyards, biocontrol, which is a presumed healthier method of controlling grapevine pathogens, has been developed. In this context, numerous bacterial and fungal potential biocontrol agents (BCAs) have been tested to control gray mold and downy mildew. As for bacteria, it has been reported that various strains of *Bacillus*, *Pseudomonas,* and *Streptomyces* significantly inhibit *B. cinerea* and *P. viticola*, both in vitro and *in planta* [4,5], by producing a wide variety of antifungal substances from their secondary metabolism, forming a biofilm that prevents the establishment of pathogenic fungi and induces plant defense responses [4,5].

Regarding fungi and yeasts, strains with different mechanisms of action from the genera *Aureobasidium*, *Fusarium*, *Rhizophagus*, and *Trichoderma* inhibit *B. cinerea* and *P. viticola* [4,5]. For example, *Fusarium* strains have the ability to inhibit both pathogens by the activation of the plant defense system and/or hyperparasitism and lytic enzyme production [6,7]. Another biocontrol agent, *T. harzianum* T39, reduces the severity of downy mildew in Pinot Noir, a susceptible grapevine cultivar, by the induction and priming of plant defense genes [8].

The activation of resistance against downy mildew and gray mold in grapevines has been demonstrated following treatments with chemical inducers, including chitosan, BTH, laminarin, and b-aminobutyric acid (BABA) [4,9,10,11]. Consequently, using plant defense stimulators is a promising strategy to control *B. cinerea* and *P. viticola*.

Regarding biocontrol oomycetes, *Pythium oligandrum*, the potential of which has been explored in many reports over the last decade, is the most widely used [12]. It naturally colonizes the rhizosphere of plants, including grapevines [13], and reduces the incidence of disease in many plants through modes of action that are either direct, i.e., mycoparasitism and antibiosis, or indirect, i.e., the induction of plant resistance [14,15,16]. Regarding grapevine diseases, *P. oligandrum* significantly reduces the necroses caused by grapevine trunk disease (GTD) pathogens in grapevine cuttings [17,18,19], and this is associated with the triggering of the jasmonate/ethylene-dependent plant defense pathways [18,20]. In addition, *P. oligandrum* and/or its elicitors activate grapevine defenses against foliar diseases, such as gray mold and powdery mildew [13,21]. For instance, Mohamed et al. [21] showed that 75% of the protection against *B. cinerea* was obtained after treatment of the grapevine root (Pinot Noir cultivar) with *P. oligandrum*, and certain plant defense genes, i.e., β 1–3 Glucanase and Stilbene Synthase, were induced in plants inoculated with *P. oligandrum* in response to *B. cinerea* attack. In another study, grapevine leaves subjected to either *P. oligandrum* culture filtrate or oligandrin experienced a reduction in powdery mildew infections due to *Erysiphe necator* [13]. QiuYan et al. [22] showed that grape resistance induced by *P. oligandrum* to control downy mildew increased when the biocontrol oomycete was combined with the fungicide dimethomorph.

A microbial biopesticide, named Polyversum^®^, based on *P. oligandrum*, is currently registered to control various plant pathogens. In raspberries, the efficacy of Polyversum^®^ against *B. cinerea* infections has been shown to be relatively similar to that obtained with a chemical treatment [23], but on grapevines, it is less effective than other resistance inducers in the control of downy mildew [24].

In this context, and considering that downy mildew and gray mold can occur on the same plants during the same harvest year, it seemed useful to study *P. oligandrum*’s ability to control these two diseases on two grapevine cultivars with different sensitivities. The other underlying idea is that the success of biocontrol measures depends on the targeted pathogen. For example, Haidar et al. [25] demonstrated that one strain of *Bacillus* (S43) efficiently inhibited *B. cinerea*, but it increased the symptoms of another pathogen, *Neofusicoccum parvum*, in grapevine cuttings. Thus, our aim was to implement a two-disease approach to assess the ability of *P. oligandrum* to protect grapevines from the necrotrophic fungus *B. cinerea* and the biotrophic oomycete *P. viticola*. A complementary objective was to decipher the molecular responses of grapevines induced by *P. oligandrum* in response to each of the two pathogens. The expression of a set of 10 defense-related genes was quantified at the leaf level using quantitative reverse-transcription polymerase chain reaction (RT-qPCR). Plant responses to each pathogen infection were then compared for the two cultivars.

## 2. Materials and Methods

### 2.1. Plant Material and Growth Conditions

Two grapevine cultivars (*Vitis vinifera* L.), Chardonnay and Cabernet Sauvignon, with different susceptibilities to two major grapevine diseases, i.e., downy mildew and gray mold, were used in this study. The Chardonnay cultivar is susceptible to *P. viticola* (the oomycete responsible for downy mildew) and *B. cinerea* (the fungus responsible for gray mold), while the Cabernet Sauvignon cultivar is less susceptible to these pathogens. One-node cuttings were placed in 350 mL pots containing Gramoflor Special soil (Gramoflor GmbH & Co., KG, Vechta, Germany) and grown in a greenhouse at a temperature of 24 °C/day and 20 °C/night, with 60% humidity and a photoperiod of 16 h of light. They were watered twice a day.

### 2.2. Microorganisms

The potential oomycete biocontrol agent *P. oligandrum*, strain Po37 [26], was isolated from grapevine roots. The oomycete inoculum was prepared by GreenCell (Saint Etienne Chomeil, France) according to the method previously described by Le Floch et al. [27]. The *P. viticola* isolate PV221 [28] used in this study was selected from the INRAE-UMR 1065 SAVE collection, Bordeaux. This strain was originally collected on *V. vinifera* in a commercial vineyard (Les Lev̀es, France) in 2014. In order to multiply the *P. viticola* isolate, droplets (15 μL) of a spore suspension (10,000 sporangia/mL) were deposited onto the abaxial face of the Cabernet Sauvignon leaves. After seven days of incubation under controlled conditions (22 °C, RH > 90%, with a 16 h light photoperiod), an inoculum of *P. viticola* sporangia was prepared by washing the freshly sporulating lesions with sterile distilled water. The sporangial suspension was then adjusted to a concentration of 10,000 sporangia/mL using a Malassez hemocytometer.

The *B. cinerea* strain CCB16 used in this study was selected from the INRAE-UMR 1065 SAVE collection, Bordeaux. It was originally collected in 2016 from naturally infected grapevine berries from the Chateau Couhins vineyard, France. The strain was maintained on malt-agar medium (MA, 20 g/L of malt from Biokar Diagnostics and 15 g/L of agar from Biokar Diagnostics, Allonne, France) at 4 °C. In order to prepare the inoculum, the *B. cinerea* strain CCB16 was grown on MA medium, incubated at 22 °C and exposed to a 12 h photoperiod using white and near-UV (UV-A at 370 nm) light (Black Light UV-A, L18 w/73, OSRAM, Munich, Germany) for 15 days [29]. This *B. cinerea* strain was incubated for 15 days at 22 °C under ultraviolet (UV) light to stimulate the sporulation. Conidia were collected with 10 mL of sterile water as described previously by Aziz et al. [30]. The suspension obtained was filtered through sterile filter paper (11 µm) to remove the mycelium. The inoculum concentration was determined by using a Malassez hemocytometer and adjusted with sterile water to 5 × 10^5^ conidia/mL.

### 2.3. Experimental Design and Plant Inoculations

For each cultivar, 60 plants were used per experiment. The plants were divided into three batches of 10 plants for the control and three batches of 10 plants for the biocontrol agent treatments. Three series of independent experiments were conducted. At the five-to-six leaf stage, the grapevine plants were inoculated with the biocontrol agent using the following protocol: 50 mL of the oomycete inoculum was applied at the collar level of each plant. The inoculum concentration was adjusted to 2 × 10^4^ oospores/mL. Fourteen days after *P. oligandrum* inoculation (14 dpi), the third or fourth leaves from the apex were collected (one leaf per plant).

The grapevine leaves were dried with filter paper, and then foliar discs of 2 cm in diameter were excised using a cork borer, as described previously by Dufour et al. [31], and deposited in Petri dishes containing Whatman paper moistened with 3 mL of sterile water. Eight discs were deposited per dish. For each cultivar, the experimental design consisted of foliar discs (i) infected with the pathogen, (ii) elicited with *P. oligandrum* and infected with the pathogen, (iii) elicited with *P. oligandrum* and non-infected with the pathogen, and (iv) non-elicited with *P. oligandrum* and non-infected with the pathogen. For the two last treatments, water droplets were deposited on the foliar discs.

On the same day (14 dpi), the plants roots were sampled to evaluate their colonization with *P. oligandrum*. For this, a plate-counting method was used, as described by Yacoub et al. [18] and Le Floch et al. [27]. Briefly, 20 root fragments per plant were sampled and deposited on the selective medium CMA-PARP (corn meal agar (a corn meal infusion of 2.0 g made with 7.0 mL of Tween 80 and 15.0 gm of Agar) with Pimaricin, Ampicillin, Rifampicin, and Pentachloronitrobenzene added) [32] and incubated at 25 °C in the dark for 10 days [18]. For each root fragment, the presence of typical *P. oligandrum* echinulated oospores was measured via an optical microscope. These echinulated oospores were identified based on Plaats-Niterink’s (1981) book [33] containing the morphological descriptions of *Pythium* species. The average frequency of *P. oligandrum* oospores was determined for each treatment.

For foliar disc *P. viticola* infection, three droplets (15 μL) of the prepared sporangia suspension (10,000 sporangia/mL) were deposited on the abaxial side. Two Petri dishes were used per treatment. The foliar discs were incubated overnight, and then the residual water was removed. Finally, the discs were incubated for seven days under controlled conditions (22 °C, RH > 90% with a 16 h light photoperiod). For the two cultivars, downy mildew development was measured according to the density of the mycelium and sporulation [34,35] at five, six, and seven days after pathogen infection. At 7 dpi, the pathogen sporangia were automatically counted in the *P. viticola* and *P. oligandrum* + *P. viticola* treatments using a particle counter, as described by Delmas et al. [36].

Regarding *B. cinerea* infection, the foliar discs were wounded on the abaxial side using a needle before pathogen infection. Then, a 5 μL drop of the prepared conidia suspension of *B. cinerea* (5 × 10^5^ conidia/mL) was deposited on the wound. For the control foliar discs, a water drop was used and five Petri dishes were used per treatment. The inoculated leaf discs were placed in a growth chamber for nine days at 25 °C with a 16 h light/8 h dark photoperiod. Gray mold development was evaluated by measuring the necrosis length three, five, six, seven, eight, and nine days after the pathogen infection [30,37]. As the overall leaf disc surface was necrosed, disease evaluation was stopped seven days after pathogen infection for the most sensible cultivar (Chardonnay).

### 2.4. RNA Extraction and RT-qPCR

In order to evaluate the grapevines’ defenses induced by *P. oligandrum*, foliar discs were collected 48 h after pathogen inoculation. For each treatment, 15 discs were collected and regrouped on three independent biological replicates. The samples were stored at –8 °C for the RT-qPCR analyses.

The RNA extraction protocol was performed according to Reid et al. [38]. After being ground in liquid nitrogen, 1 mL of an extraction buffer (300 mM Tris HCl, pH 8.0; 25 mM EDTA, 2 mM NaCl, 2% CTAB, 2% polyvinylpolypyrrolidone (PVPP), 0.05% spermidine trihydrochloride, and 2% β-mercaptoethanol added extemporaneously) that had been preheated to 65 °C was added to 200 mg of leaf powder. The mixture was stirred vigorously and incubated in a water bath at 65 °C for 10 min with regular stirring. An equal volume of chloroform/isoamyl alcohol (24:1, *v*/*v*) was added and the mixture was centrifuged at 3500× *g* for 15 min (at 4 °C). The following RNA extraction steps were conducted using a Spectrum™ Plant Total RNA Kit, according to the manufacturer’s instructions: The obtained RNA was reverse-transcribed to cDNA using M-MLV reverse transcriptase (Promega, Madison, WI, USA) following the manufacturer’s instructions.

### 2.5. Quantitative Polymerase Chain Reaction

We evaluated the expression of 10 defense-related genes using RT-qPCR. These genes encode PR proteins (*PR2*, *GLU*, and *PR5*) or are involved in the jasmonic acid (*AOS* and *JAR*), ethylene biosynthetic (*ACC* and *EIN3*) and oxylipin (*LOX2*) pathways, as well as in phenylpropanoid-related genes (*PAL* and *STS*) [31,35]. Tubuline alpha (*TUA*) and TIP41-like protein (*TIP41*) were used as the reference genes to calculate the transcripts’ relative gene expression. PCR reactions were realized in a Stratagene Mx3005P qPCR thermocycler (Agilent Technologies, Santa Clara, CA, USA) with SYBR Green dsDNA binding die, 2x MESA BLUE qPCR MasterMix Plus for SYBR^®^ Assay Low ROX (Eurogentec, Seraing, Belgium). The data were analyzed using MxPro qPCR software (Agilent Technologies) to obtain the cycle quantification number (Cq) corresponding to the fluorescence signal of the amplified DNA intersected with the background noise for each sample/gene couple. The relative expression (RE) of genes was calculated using the 2^−ΔΔCq^ method, where ΔΔCq was the ΔCq between one sample and the control. The geometric mean of the reference genes was used as an accurate normalization factor to calculate the RE levels of the genes [39].

### 2.6. Statistical Analyses

Statistical analyses were performed using the R statistical software (version R 3.1.2). For the disease evaluation experiment, data were compared at each sampling time point using the non-parametric Kruskal–Wallis test (*p* < 0.05). For the RT-qPCR experiments, principal component analysis (PCA) was carried out using the RCMD package (version 2.6-2) and the plug-in FactoMiner (version 1.7) of R statistical software. For each gene, differential gene expression was subjected to statistical analyses using non-parametric multiple comparisons with the “nparcomp” package in R software, and significant differences compared to the untreated control were determined via Tukey’s test (*p* < 0.05).

## 3. Results

### 3.1. Assessment of Grapevine Root Colonization by P. oligandrum

As shown in Figure 1, overall, the root systems of the Cabernet Sauvignon and Chardonnay plants were well colonized with *P. oligandrum* 14 days after inoculation. The percentage of *P. oligandrum* root colonization tended to be higher in the Cabernet Sauvignon plants (62%) than in the Chardonnay plants (58%). *P. oligandrum* was not detected on the roots of the control plants of the two cultivars.

### 3.2. Protection of Grapevine Leaves against B. cinerea and P. viticola

#### 3.2.1. Reduction of *P. viticola*’s Sporulation on *P. oligandrum*-Inoculated Plants

Figure 2 shows that the growth of *P. viticola* in the plants pre-treated with *P. oligandrum* before pathogen inoculation was significantly reduced on the leaves of the Cabernet Sauvignon and Chardonnay plants, at all different sampling time points. Regarding the Chardonnay plants, *P. oligandrum* significantly reduced the pathogen growth percentage when compared with the corresponding controls at 5, 6, and 7 dpi, with inhibition levels ranging from 13% to 25%. The inhibition growth of *P. viticola* stayed at the same level over time (Figure 2B).

On the Cabernet Sauvignon plants, the percentage of *P. viticola* growth leaf discs in the controls inoculated only with *P. viticola* reached 48%, 56%, and 72% at 5, 6, and 7 dpi, respectively. These values were significantly reduced to 30%, 36%, and 48% at 5, 6, and 7 dpi, respectively, on the leaf discs collected from the plants inoculated with *P. oligandrum* at the root level, before *P. viticola* inoculation (Figure 2A). The leaf discs exhibited no symptoms in the control or *P. oligandrum*-inoculated plants.

The quantification of *P. viticola* sporangia was evaluated at 7 dpi using a particle counter, and for the two cultivars, the sporulation of *P. viticola* was significantly reduced on leaf discs collected from plants pre-treated with *P. oligandrum* and then infected with *P. viticola*, in comparison to those of the controls, which were inoculated only with the pathogen (Figure 3). The observed reduction was more important on the Chardonnay leaf discs, which was approximately 40% on the Chardonnay leaves, while it was approximately 25% on the Cabernet Sauvignon leaves.

#### 3.2.2. Reduction of *B. cinerea’s* Necroses on *P. oligandrum*-Inoculated Plants

Figure 4 shows the measurement of the necroses due to *B. cinerea* on the surface of the leaf discs of Cabernet Sauvignon and Chardonnay at different sampling time points.

Regarding the controls inoculated only with *B. cinerea*, the results indicate that the severity of the pathogen attack depends on the cultivar. While the necrosis lesions of *B. cinerea* reached 1 cm at 9 dpi on the Cabernet Sauvignon leaves, these lesions were higher than 1 cm after only 7 dpi on the leaves of the Chardonnay. While the average fungal growth rate was 0.63 cm per day on the leaf discs of the Chardonnay controls inoculated with *B. cinerea* only, it was 0.47 cm on the same control of Cabernet Sauvignon. Thus, the inhibitory effect of *P. oligandrum* against *B. cinerea* depends on the grapevine cultivar.

The leaf discs of the Chardonnay plants pre-inoculated with *P. oligandrum* before inoculation with *B. cinerea* displayed significantly lower (*p* < 0.05) necrotic lesions than in those of the controls. This inhibition was approximately 35% at all sampling time points (Figure 4A). As for the leaf discs of the Cabernet Sauvignon, when the plants were pre-inoculated with *P. oligandrum*, a significant reduction in necrotic lesions due to *B. cinerea* was observed only at the late sampling time points, i.e., 8 and 9 dpi. The most important reduction was approximately 35% at 9 dpi (Figure 4B).

### 3.3. Evaluation, in the Two Cultivars, of Specific Foliar Grapevine Gene Defense Induction by P. oligandrum in Response to P. viticola and B. cinerea Infection

The *P. oligandrum* root inoculation on the grapevines’ foliar gene defenses was assessed in response, or not, to *P. viticola* or *B. cinerea* infection on the two cultivars. In the Cabernet Sauvignon and Chardonnay cultivars, the expression levels of 10 defense-related grapevine genes were evaluated two days post-inoculation with the pathogens. The gene expression levels in the control plants, not inoculated with *P. oligandrum* or any other pathogen, were used as references.

#### 3.3.1. *P. oligandrum* Induction of Grapevine Gene Defenses against *P. viticola* Attack

PCA was performed to evaluate possible separation between plant responses to the different treatments (Figure 5A and Figure 6A), and the correlation circles were examined to characterize the effect of each treatment on grapevine gene defense responses (Figure 5B and Figure 6B).

The results obtained with the Cabernet Sauvignon cultivar are shown in Figure 5A,B. The PCA eigenvalues indicate that the first two principal components explained 80% of the total data variance. As the ellipses overlapped at certain points, the grapevines’ responses for each treatment were not significantly different (Figure 5A). The correlation circles revealed that except for the gene *AOS*, all of the studied genes were well represented (Figure 5B). The position of the different genes on the correlation circle shows that all of them were associated with the *P. viticola* and *P. oligandrum* + *P. viticola* treatments. Most of these genes (*PR1*, *GLU*, *PR5*, *LOX2*, *JAR*, *PAL*, and *STS*) were expressed slightly more in those leaf discs elicited with *P. oligandrum*, followed by *P. viticola*, than in those inoculated with the pathogen only.

Regarding the Chardonnay cultivar, the plant responses for each treatment, i.e., the expression level of the 10 genes involved in the grapevine defense system, are shown in Figure 5C,D. The PCA eigenvalues indicate that 48.89% and 21.55% of the total data variance were explained by the two first principal components Dim1 and Dim2, respectively (Figure 5C). The results show that *P. oligandrum* significantly modulated the grapevines’ responses to *P. viticola* attack. As for the plants inoculated with *P. oligandrum*, the responses were not significantly different in those grapevines infected with *P. oligandrum* alone and in those with the pathogen only. The corresponding correlation circle was examined (Figure 5D) and only well-represented genes were presented. Most of these genes correlated with the grapevines’ responses to *P. oligandrum* treatment, in the presence or absence of the pathogen.

In order to better understand the effect of *P. oligandrum* on the relative expression of the studied genes, two days after *P. viticola* inoculation, Table 1A,B shows the expression level of each gene according to the individual treatments for the Cabernet Sauvignon and Chardonnay cultivars, respectively.

Regarding the Cabernet Sauvignon cultivar, the results show that all of the studied genes, except PR1, were not induced in response to *P. oligandrum* treatment (Table 1A). Compared to the control plants, only the *PR1* expression level was significantly overexpressed 16 days after *P. oligandrum* inoculation. Leaf disc inoculation with the pathogen significantly induced the expression of 40% and 80% of the studied genes in the *P. viticola* and *P. oligandrum* + *P. viticola* treatments, respectively. In the leaf discs collected from the *P. oligandrum*-treated plants that were then infected with the pathogen, only two genes involved in the signaling pathways (*AOS* and *EIN3*) were not significantly overexpressed.

As for the Chardonnay cultivar, the results are presented in Table 1B. As observed with the Cabernet Sauvignon cultivar, most of the studied genes were not significantly expressed 16 days after *P. oligandrum* inoculation, except for those genes involved in secondary metabolite biosynthesis (*PAL* and *STS*), which were slightly repressed. Following the pathogen inoculation, the expression levels of all of the studied genes (except *LOX2*) were repressed in all of the treatments, but the downregulation of these genes varied according to the treatment. The highest repression levels were observed in those leaf discs inoculated with the pathogen only, especially *PR5* (ER = 0.04), *AOS* (ER = 0.1), and *STS* (ER = 0.09).

#### 3.3.2. *P. oligandrum* Induction of Grapevine Defenses against *B. cinerea*

As per the previous study on grapevine foliar defenses after *P. viticola* inoculation, PCA and correlation circles were carried out in the same way with *B. cinerea* treatment.

For the Cabernet Sauvignon cultivar, the PCA eigenvalues indicate that the first two principal components explained 75% of the total data variance. The results show that the grapevines’ responses differed significantly according to the treatment (Figure 6A). The correlation circles of the studied genes reveal that, except for *LOX2*, all of the genes were induced in response to the pathogen attack (Figure 6B). The level of this gene expression was higher in leaf discs collected from *P. oligandrum*-inoculated plants and then infected by *B. cinerea*, than in those inoculated with the pathogen only.

Regarding the Chardonnay cultivar, the PCA eigenvalues indicate that the first two principal components, Dim1 and Dim2, explained 73% of the total data variance. Similarly to the Cabernet Sauvignon gene responses, the Chardonnay responses to *P. oligandrum* differed significantly from those of the *B. cinerea* and *P. oligandrum* + *B. cinerea* treatments (Figure 6C). Dim1, which represented 49.04%, separated specific grapevine responses into two distinct groups depending on the presence or absence of the pathogen. The correlation circles show that three of the 10 studied genes (*EIN3*, *LOX2*, and *PR1*) were associated with the plants’ responses to *P. oligandrum* (Figure 6D). However, most of the studied genes were induced in the presence of *B. cinerea* (*JAR*, *STS*, *PAL*, *GLU*, *ACC*, and *AOS*). When the plants were pretreated with *P. oligandrum*, the expression of these genes in response to *B. cinerea* infection was higher than those observed in leaves infected with the pathogen only.

In order to go further with the analyses of the grapevines’ gene expression in response to *B. cinerea* and *P. oligandrum*, Table 1C,D shows the expression level of each gene for each treatment. For the Cabernet Sauvignon cultivar, the majority of the 10 studied genes were not highly modulated in the leaf discs of those plants inoculated with *P. oligandrum* alone, except for *PR5*, which was four times more induced. Following inoculation with B. cinerea, all of these genes were induced more, except *LOX2*, which was significantly downregulated. The gene that showed the highest level of expression in the presence of the pathogen (100 times more expressed than the control) was the one involved in the jasmonate synthesis pathway, *AOS*. Overall, all of the studied genes were induced more in the leaf discs of those plants inoculated with *P. oligandrum* and *B. cinerea* than in those of the plants inoculated with B. cinerea only. Thus, inoculation of the plant roots with *P. oligandrum* strongly induced the expression of the studied genes, especially *PR5* and *AOS*, for which the level of expression was multiplied by five in the leaves inoculated with *P. oligandrum* and then infected with B. cinerea compared to those inoculated with the pathogen only. As for the genes involved in the synthesis of secondary metabolites (i.e., *PAL* and *STS*), their relative expression levels were twice as high in the presence of *P. oligandrum* and *B. cinerea*.

Regarding the Chardonnay cultivar, the results show that in presence of the pathogen, most of the 10 studied genes were strongly induced (Table 1D). As an example, the genes involved in secondary metabolite synthesis (i.e., *PAL* and *STS*), signaling pathways (i.e., *AOS*, *ACC*), and PR proteins (i.e., *PR1* and *GLU*) were at least three times more overexpressed in plants inoculated with *B. cinerea* than in the controls. In the Chardonnay plants, the gene that showed the highest expression level (5890 times more than in the control plants), in response to pathogen infection, was the *AOS* gene involved in jasmonate synthesis. The expression of the *PR1*, *PR5*, *AOS*, and *LOX2* genes were induced more in the leaf discs of those plants inoculated with *P. oligandrum* and *B. cinerea* than in those plants inoculated with the pathogen only. Similarly to what was observed for the Cabernet Sauvignon cultivar, the level of expression of the *AOS* gene was more strongly induced in those plants inoculated with P. oligandrum and *B. cinerea* (relative expression = 8009) than in those infected by the pathogen only (relative expression = 5890).

## 4. Discussion

Based on the hypothesis that *P. oligandrum* enhances grapevine resistance, we implemented a two-disease approach to decipher the induced plant responses when two major grapevine leaf and berry pathogens attack young vines. We selected two pathogens with different trophic behaviors, i.e., *P. viticola* (a necrotrophic oomycete) and *B. cinerea* (a biotrophic fungus), and we used two cultivars, i.e., Cabernet Sauvignon and Chardonnay, with different susceptibilities to both diseases.

With root colonization being an absolute pre-requisite for a biocontrol agent to protect plants against pathogenic attacks [40,41], we first checked this point. Under our experimental conditions, we confirmed that *P. oligandrum* effectively colonizes the roots of Cabernet Sauvignon and Chardonnay, two grapevine cultivars among the most widely planted in the world. Then, before deciphering the induced resistance, we estimated the level of biocontrol protection. Unlike Haidar et al. [25], who observed in another two-pathogen approach that certain bacteria used as biocontrol agents inhibit *B. cinerea* infection but favor that of another pathogen, namely *Neofusiccocum parvum*, in our experiment, both pathogenic infections were reduced when *P. oligandrum* colonized the roots of the vines. For *P. viticola*, the pathogen growth was reduced by 35% regardless of the cultivar used, while there was a 30–35% reduction in the development of *B. cinerea* necrosis on the leaves of the Chardonnay and Cabernet Sauvignon.

With respect to *P. viticola*, this result is consistent with what has previously been documented by Taibi et al. [24] on the ability of the biocontrol product Polyversum (based on *P. oligandrum*) to control *P. viticola* on the grapevine cultivar Barbera. As for *B. cinerea*, the protection we obtained was 30–35%, and the necroses of *B. cinerea* in the Chardonnay leaf discs were higher than those measured on the Cabernet Sauvignon leaf discs. This result is in accordance with [42,43], who showed that the Chardonnay cultivar is more susceptible to gray mold than Cabernet Sauvignon. In a previous experiment with *P. oligandrum*, Mohamed et al. [21] obtained 75% protection on the Pinot Noir cultivar, known for its high susceptibility to *B. cinerea*. The differences in protection (35% vs. 75%) between our study and that of Mohamed et al. [21] are certainly linked to the different experimental conditions used.

In the literature, further to plant inoculation with potential BCAs, the induction of resistance was reported against *P. viticola* and *B. cinerea* attack. For instance, Lakkis et al. [37] indicated that the bacterial strain *Pseudomonas fluorescens* PTA-CT2 induced systemic resistance (ISR) against *P. viticola* and *B. cinerea* by priming various defensive pathways. Among the fungi, various *Trhichoderma* species were able to induce a priming-type systemic defensive response against *B. cinerea* through the inhibition of ROS production [44]. Similarly for *P. viticola*, several fungal species with biocontrol potential induced plant resistance against this pathogen [5]. Therefore, the grapevine resistance induced by the oomycete *P. oligandrum* observed in this study is consistent with this literature [4,5]. However, our results also showed that the relative expression levels of the genes depended on the two cultivars and the two pathogens.

Regarding the plant interaction with the microorganisms we used, it was first shown that *P. oligandrum* inoculation alone did not induce any consistent change in the basal expression of the defense genes in the Chardonnay and Cabernet Sauvignon cultivars. Therefore, *P. oligandrum* presumably primed the plants to enhance their resistance when a subsequent pathogenic attack occurred. This result is in agreement with previous studies on *P. oligandrum* and other BCA plant inoculations alone, showing that they were not associated with a significant induction of plant defenses [4,5].

When the leaf discs were challenged with the pathogens, the grapevines’ defenses were activated, though they varied depending on each pathogen/cultivar combination. In another study, *Solanum lycopersicum* L. cv. Micro-Tom seeds were coated with *P. oligandrum* strains, and then the seedlings were exposed to fungal pathogens, i.e., *Alternaria brassicicola* or *Verticillium albo-atrum* [45]. Their results depended on the plant species, but also, as is the case in our study, on the cultivars and type of pathogens.

Thus, a difference in the relative expression of the studied genes in response to each pathogen (*B. cinerea* and *P. viticola*) was observed in our study. This could be attributed to the lifestyles of the two pathogens, i.e., biotrophic or necrotrophic, which are linked to the activation of the specific plant metabolic pathways. Indeed, the mechanisms of plant defense against biotrophic pathogens described in the literature are associated with responses regulated by the salicylic-acid-dependent pathway, whereas for the necrotrophic pathogens, there was a different set of defense responses activated by jasmonic acid and ethylene signaling [46,47]. Here, in response to *P. viticola* infection, in Cabernet Sauvignon, the genes from the jasmonate and ethylene pathways were induced more than in the more susceptible Chardonnay cultivar. This result is in accordance with several authors [46,47,48], who reported that *P. viticola* infection results in an induction of the genes involved in jasmonate and ethylene biosynthesis, which is more important in resistant than susceptible cultivars.

A different result was observed with *B. cinerea*; the relative expression levels of the studied genes involved in PR protein, ethylene/jasmonate, and secondary metabolite biosynthesis were higher in the susceptible Chardonnay cultivar than in the less-susceptible Cabernet Sauvignon cultivar. In another experiment with a different plant/pathogen system, the JA-dependent signaling pathway was reported to be required for the induction of tomato resistance by *P. oligandrum* against *Ralstonia solanacearum* [49].

Regarding the studied genes, we showed that the *AOS* gene involved in the jasmonate pathway was strongly induced in response to *B. cinerea* compared to *P. viticola* infection in leaf discs primed, or not, with *P. oligandrum*. The same kind of result was observed with the *AOS* gene in a previous study involving *P. oligandrum* control of the GTD pathogen *P. chlamydospora* [20]. Consequently, the *AOS* gene could be an interesting marker for the grapevine protection induced by *P. oligandrum* against certain grapevine pathogens.

## 5. Conclusions

The two-pathogen approach used in this paper provided evidence that the strain of *P. oligandrum* we used has real potential to control attacks of both *P. viticola* and *B. cinerea* on grapevines. The different levels of defense against *B. cinerea* and *P. viticola* could explain the difference in susceptibility of the two grapevine cultivars (Chardonnay and Cabernet Sauvignon) to these two pathogens. Moreover, as the efficacy of BCA products based on *P. oligandrum*, e.g., Polyversum^®^, against *P. viticola* and *B. cinerea* is variable in vineyards, a solution to improve and secure the protection offered by *P. oligandrum* could be the use of mixtures of *P. oligandrum* strains, such as the one used in this paper. This mixture of *P. oligandrum* strains could then be integrated into agroecological strategies for grapevine protection. Further studies, but under viticultural conditions, should be carried out.

## Figures and Tables

**Figure 1 jof-09-00511-f001:**
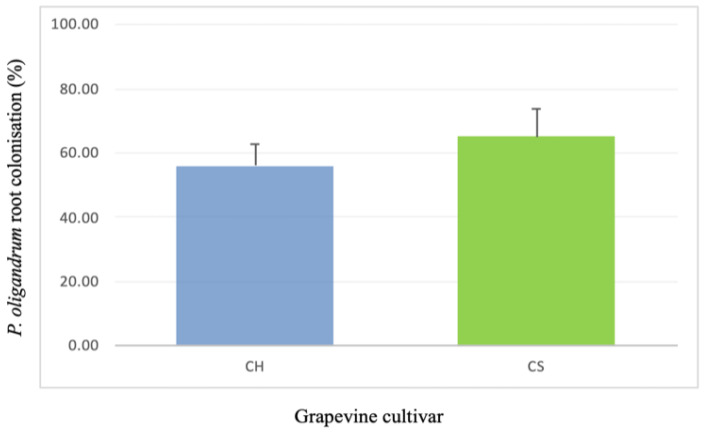
Root colonization of grapevines by *P. oligandrum*, assessed via plate counting, on two different cultivars 14 days after oomycete inoculation. CH: Chardonnay cultivar; CS: Cabernet Sauvignon cultivar. No significant differences were detected between the cultivars.

**Figure 2 jof-09-00511-f002:**
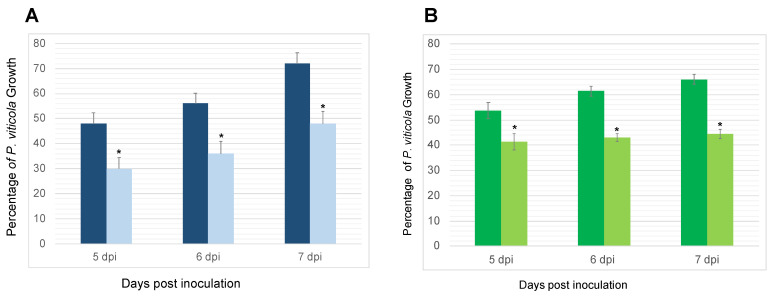
*P. oligandrum* reduced *P. viticola* growth on the leaf discs of the two grapevine cultivars: Cabernet Sauvignon (**A**) and Chardonnay (**B**), at different sampling time points. For each cultivar, the dark-colored bars correspond to *P. viticola* treatment and the light-colored ones correspond to *P. oligandrum* + *P. viticola* treatment. dpi: Days post-inoculation with the pathogen. Comparisons of statistical significance were made at each sampling time point. Data are the means from three independent experiments; each experiment was performed with 16 leaves for each condition, and the vertical bars indicate the standard error. * Significantly different between the two treatments at *p* < 0.05 (Kruskal–Wallis test).

**Figure 3 jof-09-00511-f003:**
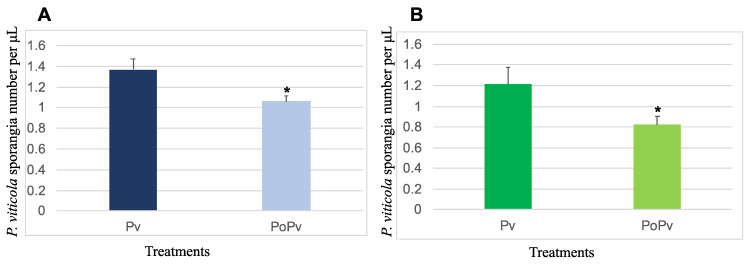
*P. oligandrum* reduced *P. viticola* sporulation in the leaf discs of the two grapevine cultivars: Cabernet Sauvignon (**A**) and Chardonnay (**B**) at 7 days post-inoculation by the pathogen. Pv and PoPv correspond to the *P. viticola* and *P. oligandrum* + *P. viticola* treatments, respectively. * Significantly different between the two treatments at *p* < 0.05 (Kruskal–Wallis test).

**Figure 4 jof-09-00511-f004:**
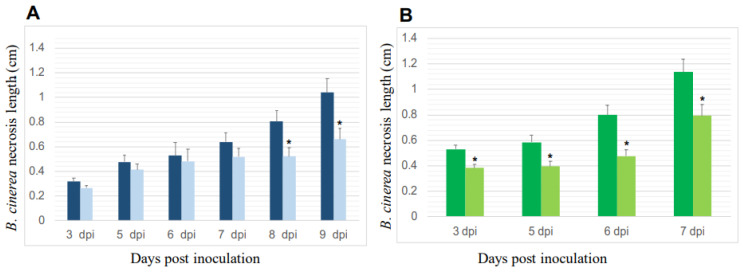
*P. oligandrum* reduced *B. cinerea* necroses in the leaf discs of two grapevine cultivars: Cabernet Sauvignon (**A**) and Chardonnay (**B**), at different sampling time points. For each cultivar, the dark-colored bars correspond to *B. cinerea* treatment and the light-colored ones correspond to *P. oligandrum* + *B. cinerea* treatment. dpi: Days post-inoculation with the pathogen. Comparisons of statistical significance were made at each sampling time point. Data are the means from three independent experiments; each experiment was performed with 30 leaves for each condition, and the vertical bars indicate the standard error. * Significantly different between the two treatments at *p* < 0.05 (Kruskal–Wallis test).

**Figure 5 jof-09-00511-f005:**
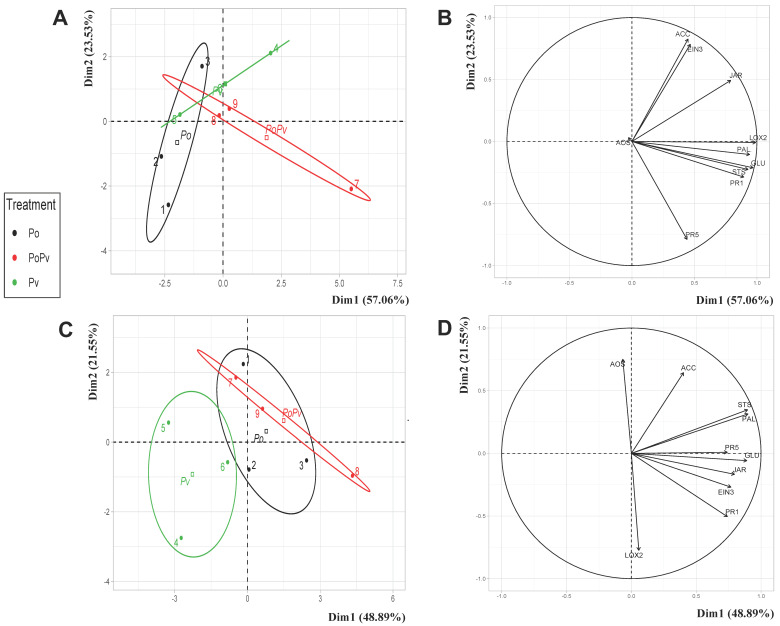
Principal component analysis of specific responses to the *P. oligandrum*, *P. viticola*, and *P. oligandrum* + *P. viticola* treatments (relative expression levels of the 10 genes involved in plant defenses), at 48 h after inoculation, or not, with the pathogen on two grapevine cultivars: Cabernet Sauvignon (**A**) and Chardonnay (**C**). Distribution in the correlation circles of the relative expression of the 10 genes studied on the Cabernet sauvignon (**B**) and Chardonnay cultivars (**D**). The gene expression of the control plants was used as the reference to calculate the relative expression. Ellipsoids represent the centers of the factors with 95% confidence. The different groups are indicated by different colors. Po: *P. oligandrum*; Pv: *P. viticola*; PoPv: *P. oligandrum* + *P. viticola*.

**Figure 6 jof-09-00511-f006:**
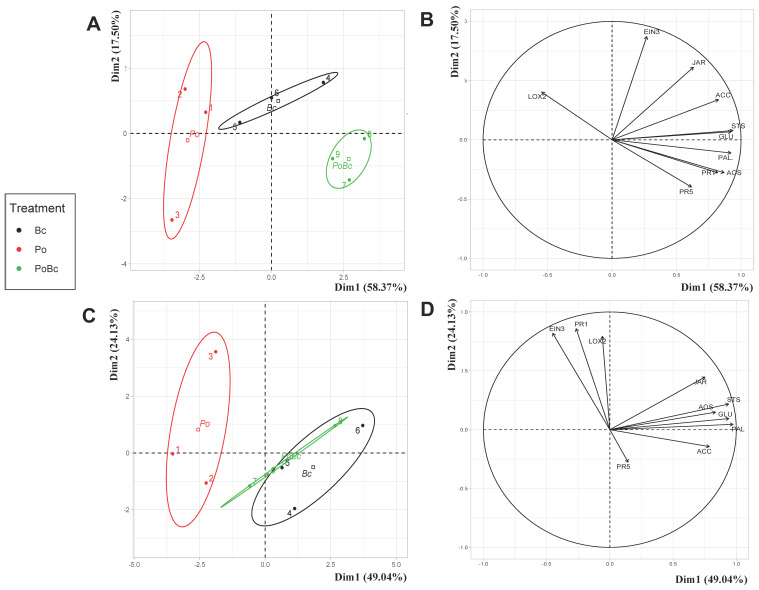
Principal component analysis of specific grapevine responses to the *P. oligandrum*, *B. cinerea*, and *P. oligandrum* + *B. cinerea* treatments (relative expression levels of the 10 genes involved in plant defenses), at 48 h after inoculation, or not, with the pathogen, on two grapevine cultivars: Cabernet Sauvignon (**A**) and Chardonnay (**C**). Distribution in the correlation circles of the relative expression of the 10 genes studied on the Cabernet Sauvignon (**B**) and Chardonnay cultivars (**D**). The gene expression of the control plants was used as the reference to calculate the relative expression. Ellipsoids represent the centers of factors with 95% confidence. The different groups are indicated by different colors. Po = *P. oligandrum*; Bc, *B. cinerea;* PoBc: *P. oligandrum* + *B. cinerea*.

**Table 1 jof-09-00511-t001:** *P. oligandrum* primed the differential expression of defense-related genes in the leaf discs of Cabernet Sauvignon (**A**,**C**) and Chardonnay (**B**,**D**) after *P. viticola* (**A**,**B**) and *B. cinerea* (**C**,**D**) inoculation. The gene expression of the control plants was used as the reference to calculate the relative expression. Columns represent the treatments (Po = *P. oligandrum*; Pv: *P. viticola*; PoPv: *P. oligandrum* + *P. viticola*; Bc: *B. cinerea*; PoBc: *P. oligandrum* + *B. cinerea*), and each line corresponds to one gene, represented by a single row of boxes. The color scale bars represent the ratio values corresponding to the mean of three independent samples. Upregulated genes are shown in shades of red, with relative expression levels greater than three in bright red. Downregulated genes are shown in shades of grey, with relative expression levels less than 0.1 in dark grey. Numbers in boxes represent significant changes (*p* < 0.05; Kurskal–Wallis test) in gene expression compared to the control.

A	Cabernet Sauvignon	B	Chardonnay	
Po	Pv	PoPv	Po	Pv	PoPv
** *PR1* **	**1.42**	**2.24**	**3.01**	** *PR1* **			**0.83**	
** *GLU* **			**1.52**	** *GLU* **			**0.83**
** *PR5* **			**2.81**	** *PR5* **		**0.04**	**0.39**
** *LOX2* **	**0.62**		**1.6**	** *LOX2* **			**5.08**
** *AOS* **		**3.48**		** *AOS* **		**0.1**	
** *JAR* **		**1.63**	**1.71**	** *JAR* **	**1.25**			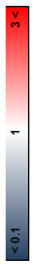
** *PAL* **		**1.45**	**2.08**	** *PAL* **	**0.48**	**0.26**	**0.57**
** *STS* **			**2.11**	** *STS* **	**0.36**	**0.09**	**0.53**
** *ACC* **			**1.16**	** *ACC* **		**0.86**	
** *EIN3* **				** *EIN3* **		**0.74**	
**C**	**Cabernet Sauvignon**	**D**	**Chardonnay**
**Po**	**Bc**	**PoBc**	**Po**	**Bc**	**PoBc**
** *PR1* **	**0.88**		**1.4**	** *PR1* **	**1.50**	**1.9**	
** *GLU* **	**0.71**	**1.34**	**1.43**	** *GLU* **	**1.56**	**7.81**	**6.97**
** *PR5* **	**3.56**	**5.36**	**26.33**	** *PR5* **	**1.43**	**5.03**	**7.15**
** *LOX2* **		**0.23**	**0.41**	** *LOX2* **	**1.42**		**1.58**
** *AOS* **		**133.97**	**673.38**	** *AOS* **	**2.00**	**5890**	**8009.14**	
** *JAR* **				** *JAR* **	**1.27**	**1.7**	**1.29**
** *PAL* **		**2.96**	**6.86**	** *PAL* **	**0.72**	**6.62**	**4.98**
** *STS* **		**2.53**	**4.51**	** *STS* **	**0.97**	**6.84**	**5.34**
** *ACC* **	**0.49**	**1.69**	**1.51**	** *ACC* **	**1.45**	**2.99**	**2.37**	
** *EIN3* **				** *EIN3* **	**1.37**	**0.87**	**0.77**

## Data Availability

Not applicable.

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
