# Peer review of "Deciphering Plant-Induced Responses toward Botrytis cinerea and Plasmopara viticola Attacks in Two Grapevine Cultivars Colonized by the Root Biocontrol Oomycete, Pythium oligandrum"

_jof, 2023, doi:10.3390/jof9050511_

Round 1
Reviewer 1 Report
Comments are included in the attached file.

Author Response
We would like to thank the reviewers for their helpful comments that were taken in account and integrated in the revised manuscript.
REVIEWER 1
All reviewer comments included in the attached file have been considered and incorporated into the revised manuscript.

Reviewer 2 Report
This paper characterize effect of P. oligandrum on grapevine defense mechanisms in combating gray mold and downy mildew causing agents. The aim of the study is not innovative enough because there are several publications on the chosen subject, however in addition to that there are several general improvements to the manuscript that are given below in addition to those given in the form of comments in pdf file. The MS was reviewed thoroughly and the following observations were made:
1. English language needs to be improved since it is not on a level expected for scientific publication. Authors are suggested to use English proofing service or to ask a colleague who is native English speaker to do proof reading.
2. Introduction is too wide, not focused and does not fully support the aim of the research.
3. All isolated used in this research (both biocontrol agent and pathogenic isolates) are not references. There is no trace that they belong to the stated species, let alone strain. Authors should include reference or detailed morphological and molecular analysis of used microorganisms.
4. Citations in M&M are incomplete and described methods are not supported with references.
5. All other suggestions are given in the manuscript.

Author Response
We would like to thank the reviewers for their helpful comments that were taken in account and integrated in the revised manuscript.
REVIEWER 2
- English language needs to be improved since it is not on a level expected for scientific publication. Authors are suggested to use English proofing service or to ask a colleague who is native English speaker to do proof reading.
The English language have been revised and improved throughout the manuscript.
- Introduction is too wide, not focused and does not fully support the aim of the research.
The introduction has been largely modified in accordance with the reviewer’s comments. This section is now more focused on our research topic to support the prupose of the implemented experiments. Similarly, the discussion has been largely modified in accordance with the reviewer’s comments.
- All isolated used in this research (both biocontrol agent and pathogenic isolates) are not references. There is no trace that they belong to the stated species, let alone strain. Authors should include reference or detailed morphological and molecular analysis of used microorganisms.
Each of the microorganisms used in this paper have been previously identified and the whole genomes of Pythium oligandrum and Plasmopara viticola were published in the scientific literature. These references have been added in the manuscript.
These references are:
- For the Pythium oligandrum strain: Po37 (Berger et al 2016)
Berger, H.; Yacoub, A.; Gerbore, J.; Grizard, D.; Rey, P.; Sessitsch, A.; Compant, S. Draft Genome Sequence of Biocontrol Agent Pythium oligandrum Strain Po37, an Oomycota. Genome. Announc 2016, 4, e00215-16, doi:10.1128/genomeA.00215-16.
- For the Plasmopara viticola strain: PV221 (Dussert et al 2016)
Dussert, Y.; Gouzy, J.; Richart-Cervera, S.; Mazet, I.D.; Delière, L.; Couture, C.; Legrand, L.; Piron, M.-C.; Mestre, P.; Delmotte, F. Draft Genome Sequence of Plasmopara Viticola, the Grapevine Downy Mildew Pathogen. Genome. Announc 2016, 4, e00987-16, doi:10.1128/genomeA.00987-16.
- For the Botrytis cinera strain: CCB16 (Bellée et al 2016)
Bellée, A.; Cluzet, S.; Dufour, M.-C.; Mérillon, J.-M.; Corio-Costet, M.-F. Comparison of the Impact of Two Molecules on Plant Defense and on Efficacy against Botrytis Cinerea in the Vineyard: A Plant Defense Inducer (Benzothiadiazole) and a Fungicide (Pyrimethanil). J. Agric. Food. Chem. 2018, 66, 3338–3350, doi:10.1021/acs.jafc.7b05725.
- Citations in M&M are incomplete and described methods are not supported with references.
Citations in the M&Ms have been modified and other comments have been addressed and incorporated into the revised manuscript.
- All other suggestions are given in the manuscript.
All reviewer comments included in the attached file have been considered and incorporated into the revised manuscript.

Round 2
Reviewer 2 Report
This is the second time I am doing revision of this manuscript which presents testing of potential biocontrol agent Pythium oligandrum in control of downy mildew and gray mold in vineyards including testing MoA pathways molecularly. Even though this manuscript is largely improved, it still lacks professional language proofing. Authors are requested to engage service which can improve readability of this article. MDPI has this kind of service as I recall.
Minor changes are given in the manuscript.
Author Response
REVIEWER 2
Thank you very much for your comments. We would like to inform you that the manuscript was checked for English with MDPI's English editing service.
